# Orthostatic Challenge-Induced Coagulation Activation in Young and Older Persons

**DOI:** 10.3390/biomedicines10112769

**Published:** 2022-10-31

**Authors:** Axel Schlagenhauf, Bianca Steuber, Markus Kneihsl, Thomas Gattringer, Martin Koestenberger, Melina Tsiountsioura, Tobias Ziegler, Erwin Tafeit, Margret Paar, Willibald Wonisch, Thomas Wagner, Andreas Rössler, James Elvis Waha, Gerhard Cvirn, Nandu Goswami

**Affiliations:** 1Department of Paediatrics and Adolescent Medicine, Division of General Paediatrics, Medical University of Graz, 8036 Graz, Austria; 2Otto Loewi Research Centre, Division of Physiology, Medical University of Graz, 8010 Graz, Austria; 3Department of Neurology, Medical University of Graz, 8036 Graz, Austria; 4Otto Loewi Research Centre, Division of Medicinal Chemistry, Medical University of Graz, 8010 Graz, Austria; 5Department of Blood Group Serology and Transfusion Medicine, Medical University of Graz, 8036 Graz, Austria; 6General, Visceral and Transplant Surgery, Department of Surgery, Medical University of Graz, 8036 Graz, Austria

**Keywords:** aging, clot formation, orthostatic challenge, thrombin generation, thrombelastometry, thrombosis

## Abstract

The incidence of thrombosis increases with aging. We investigated the coagulatory/haemostatic system across the ages and tested the hypothesis that older persons have a hypercoagulable state compared to younger persons at rest, and that standing up (orthostasis) leads to greater changes in coagulation in older persons. In total, 22 older and 20 young participants performed a 6 min sit-to-stand test (orthostatic challenge). Blood was collected prior to and at the end of standing and haemostatic profiling was performed via thrombelastometry (TEM), calibrated automated thrombogram (CAT) and standard coagulation assays. At baseline, three CAT-derived values indicated enhanced capability to generate thrombin in older participants. However, other measured parameters did not suggest a hypercoagulable state in older participants: prolonged TEM-derived coagulation times (295 vs. 209 s, medians, *p* = 0.0025) and prothrombin times (103 vs. 114%, medians, *p* = 0.0087), as well as lower TF levels (440 vs. 672 pg/mL, medians, *p* = 0.0245) and higher t-PA levels (7.3 vs. 3.8 ng/mL, medians, *p* = 0.0002), indicative of enhanced fibrinolytic capability, were seen. Younger participants were more sensitive to the orthostatic challenge: CAT-derived endogenous thrombin potentials (ETPs) were only increased in the young (1337 to 1350 nM.min, medians, *p* = 0.0264) and shortening of PTs was significantly higher in the young vs. older participants (*p* = 0.0242). Our data suggest that the increased thrombosis propensity in older persons is not primarily attributable to a hyperactive coagulation cascade but may be due to other pathologies associated with aging.

## 1. Introduction

The incidence of arterial or venous thrombosis has been reported to increase with increasing age [1,2]. Several factors have been suggested to contribute to the increased thrombotic propensity in older persons: increased coagulation proteins without proportional increases in anticoagulant factors, enhanced platelet reactivity, molecular and anatomic changes in vessel walls and impairment of fibrinolytic activity [3,4]. Increased platelet activity with age has been shown to correlate with a higher content of platelet phospholipids, increased platelet transmembrane signalling or second messenger accumulation [5]. Concomitantly, shortening of bleeding time with age has been shown [6,7].

Additional factors such as major surgery, malignant disease, as well as frequent bedrest confinement due to COVID-19, chronic diseases and post-fall may increase the risk of thrombosis, especially in older persons [8,9,10,11,12].

Despite elevated levels of several coagulation markers that are seen in older persons, many older persons do not develop clinical thrombotic events [8,13]. A possible underlying mechanism has been shown recently. Kumar et al. have shown that DNase 1 limits thrombin generation and protects from venous thrombosis during aging in mice and humans [14]. 

In this study, we investigated whether the haemostatic system of older persons is actually in a hypercoagulable state and whether an orthostatic challenge, induced by a sit-to-stand test, alters the coagulation more in older participants as compared to younger participants. We hypothesised that older participants have a greater tendency to thrombose during orthostatic loading due to age-associated changes in Virchow’s triad and shear stress induced in the vessels during standing up. We tested this hypothesis using a well-established mild coagulation activation protocol (sit-to-stand test) in 22 older and 20 younger participants [15]. It has been shown previously that a sit-to-stand test is a suitable method to induce orthostatic challenge followed by coagulation activation [16]. Blood was collected prior (baseline values) and immediately after a 6 min sit-to-stand test; pre- and post-standing haemostatic data were compared across the two age groups. Haemostatic profiling was carried out in both whole blood (WB) and platelet poor plasma (PPP) samples. Thrombelastometry (TEM) was used to monitor the clot formation process. Thrombin generation was assessed by means of calibrated automated thrombography (CAT), and by the determination of plasma levels of Prothrombin fragment 1 + 2 (F1 + 2) and Thrombin-antithrombin complexes (TAT). For further assessment of orthostatic challenge-induced coagulation activation, we measured prothrombin times (PT), activated partial thromboplastin times (APTT), plasma activities of FII, FVII, FVIII, protein C (PC), and protein S (PS). Orthostatic challenge-induced vessel damage was assessed by measuring plasma levels of TF and of tissue plasminogen activator (tPA). As standing up has been shown to be associated with haemoconcentration—due to the transfer of intravascular fluid from the blood to the surrounding tissue [17]—we evaluated haematocrit values pre- and post-standing in both groups of participants in order to compare the respective microvascular permeabilities.

## 2. Materials and Methods

### 2.1. Participants

In total, 22 healthy older participants and 20 healthy younger participants performed a sit-to-stand test. The participants’ characteristics are shown in Table 1.

### 2.2. Experimental Design

After 5 min of sitting still, a blood sample (baseline) was collected from the antecubital vein. Subsequently, the participants were assisted into the upright position for 6 min and upon sitting down again another blood sample was then withdrawn from the vein (post-standing sample). While standing, the participants kept their eyes open and did not alter foot placement (as changes in leg muscle pump are known to influence venous return) [18]. The investigations were carried out in a room with minimal ambient noise, room temperature maintained between 23–25 °C and between 7 and 11 a.m. The Ethics Committee of the Medical University of Graz, Austria approved this study (EK-Nr. 25-551 ex 12/13). The participants provided informed written consent before taking part in the study.

### 2.3. Blood Sampling

Baseline samples: 9 mL of venous blood was collected into Vacuette^®^ tubes (Greiner Bio-one GmbH, Kremsmünster, Austria). The tubes contained 3.8% sodium citrate. Haematocrit, platelet counts and TEM measurements were performed in citrated WB samples. Subsequently, the remaining WB underwent centrifugation at 500 g for 15 min in order to prepare PPP samples. The remaining measurements were performed in PPP samples. 

Post-standing samples: after the participants spent 6 min in an upright position, venous blood samples were withdrawn and processed as described for the baseline samples.

### 2.4. Thrombelastometry Assay

We used the thrombelastometry coagulation analyser (ROTEM 05, Matel Medizintechnik, Hausmannstätten, Austria) to monitor the clot formation process. Following measurands were provided: coagulation time (CT), indicating the period of time from adding trigger to initial fibrin fibres formation; clot formation time (CFT), the period of time taken until the amplitude reaches 20 mm; maximum clot firmness (MCF), reflecting clot stability; and alpha angle, indicating the rate of fibrin built-up and cross-linking. The volume of blood samples was 340 µL and clot formation was started by addition of 40 µL of “trigger solution” (containing 0.35 pmol/L TF and 3 mmol/L CaCl_2_, final concentration) to 300 µL of citrated WB, as described previously [19]. CTs and CFTs thus measured are dependent on activities of factors II, V, VII, and X; MCF is dependent on fibrinogen levels and platelet counts; VWF, F XIII and tissue plasminogen activator inhibitor have no influence on TEM [20].

### 2.5. Thrombin Generation Measurements Using Calibrated Automated Thrombography (CAT)

CAT obtained from Thrombinoscope BV (Maastricht, the Netherlands) was applied to evaluate thrombin generation curves and has been described in detail previously [21]. The PPP was centrifuged at 2600× *g* immediately before it was used for CAT measurements. Then, plasma was carefully aliquoted to the 96-well plate, leaving 100 µL at the bottom of the vial. We developed and validated these procedures years ago, when we were studying the impact of microparticles on thrombin generation [22,23]. With our centrifugation scheme, larger cell debris is removed, but the smaller fraction of microparticles remain in the sample. However, the method remains unaffected by microparticles when an excess of 4 µM phospholipids and 5 pM TF are added exogenously to the sample. This double centrifugation procedure has also been employed by H.C. Hemker, who developed the assay. We refrained from ultrafiltration because we observed substantial contact activation that would bias the results. The following laboratory measurands were evaluated: the time until the thrombin burst; lag time (LT); endogenous thrombin potential (ETP) and peak height (Peak), both indicating the amount of thrombin being generated; time-to-peak (ttPeak); VELINDEX [peak thrombin/(peak time–lag time)], the peak rate of thrombin formation; StartTail, the time from which on no free thrombin is measurable. Low amounts of trigger (5 pmol/L of TF final concentration) are used allowing sensitive detection of thrombin formation. The CAT assay is influenced by all intrinsic and extrinsic coagulation factors (except for F XII and F XIII) and by the inhibitors antithrombin, tissue factor pathway inhibitor and alpha2-macroglobulin. PC, PS and fibrinolytic factors have no influence on the CAT assay [21].

### 2.6. Standard Laboratory Tests

Determinations of PTs (expressed as % of normal), APTTs as well as of plasma activities of FII, FVII, FVIII, PC- and PS-activity were performed on an ACL Top 350 (Werfen, Germany) using the PT reagent ReadiPlasTin, the APTT reagent SynthASil and HemosIL reagents for all specialised clotting assays (all from Werfen, Germany).

ELISA kits from Behring Diagnostics (Marburg, Germany) were used in order to determine plasma levels of F1 + 2 and TAT complexes. TF was determined by using the assay ACTICHROM Tissue Factor ELISA kit. TPA was determined by using the assay IMUBIND tPA ELISA kit. Both ELISAs were from American Diagnostica (Pfungstadt, Germany).

### 2.7. Haematocrit and Blood Cell Counts

Sysmex KX-21 N Automated Haematology Analyzer from Sysmex (Lincolnshire, IL, USA) was used to evaluate haematocrit and blood cell counts.

### 2.8. Statistics

The GraphPad 8.0 Prism package was used for statistical evaluation. Differences between mean values of continuous variables (age, BMI) were tested for statistical significance by the Mann–Whitney test, frequencies of all other (categorical) variables were compared between young and older participants by Fisher’s exact test (Table 1) [24]. Differences of baseline levels between young and older participants were tested for statistical significance by means of the Mann–Whitney test (Table 2). Differences between post-standing and baseline levels were tested for significance by means of the Wilcoxon matched pairs signed-rank test (Table 3 and Table 4). Differences of coagulation value changes due to orthostatic challenge between young and older participants were tested for significance by means of the Mann–Whitney test (Figure 1). *p* values less than 0.05 were considered statistically significant.

## 3. Results

Detailed anthropometric measurements of the participants are listed (Table 1). Age and BMI were significantly higher in the older compared with the younger participants (*p* = 0.0001 and *p* = 0.0050, respectively). Most values were similar in younger and older participants. The number of participants with dyslipidaemia or arterial hypertension were markedly higher in the older compared with the younger ones (*p* = 0.0220 and *p* = 0.0049, respectively). None of the participants enrolled in this study experienced previous vascular events such as venous thromboembolism, stroke or myocardial infarction.

### 3.1. Baseline Levels of Coagulation Values in Young vs. Older Participants

Most of the baseline coagulation values were similar in young and older participants (Table 2). Interestingly, CTs (evaluated by means of TEM) as well as PTs were significantly prolonged in older participants as compared to younger ones. Three CAT values indicated an enhanced capability of older participants to generate thrombin: the (thrombin) peak and the velocity of thrombin formation (VELINDEX) were significantly higher in older as compared with younger participants. Commensurably, the time until maximum thrombin formation was significantly shorter in older participants. Moreover, PS and TF plasma levels were significantly higher and tPA plasma levels were significantly lower in young as compared with older participants (Table 2).

### 3.2. Effects of Orthostatic Challenge on Thrombelastometry Values in Young vs. Older Subjects

No effect of orthostatic challenge on TEM values was observed in young (Table 3) as well as in older participants (Table 4). 

### 3.3. Effects of Orthostatic Challenge on CAT Values in Young vs. Older Subjects

In younger participants, the ETP was significantly increased due to the orthostatic loading (Table 3), while all other CAT values remained unaltered. No effect of orthostatic challenge on CAT values was observed in older participants (Table 4).

### 3.4. Effects of Orthostatic Challenge on Standard Coagulation Times in Young vs. Older Participants

PTs were significantly shortened by orthostatic loading in both young (Table 3) and older participants (Table 4). The shortening of PTs (difference between post-standing and baseline) was significantly higher in the young as compared with older participants (Figure 1A; *p* = 0.0242). APTTs were significantly shortened to similar extents in both group of participants (Figure 1B).

### 3.5. Effects of Orthostatic Challenge on Pro- and Anti-Coagulant Factors in Young vs. Older Participants

In young participants, FVIII plasma levels were significantly higher in the post-standing PPP samples (Table 3) as compared with baseline. The extent of this change was greater than the haemoconcentration effect associated with the sit-to-stand test (~4.3% change in plasma volume). In older participants, FVIII plasma levels were not affected by orthostatic challenge (Table 4). The concentrations of the other procoagulant factors, as well as that of the anticoagulant factors PC and PS, remained unaltered by orthostatic challenge in both young (Table 3) and older participants (Table 4).

### 3.6. Effects of Orthostatic Challenge on Thrombin Generation Markers in Young vs. Older Participants

Orthostatic challenge caused markedly increased post-standing plasma levels of F1 + 2 and TAT in younger (Table 3) as well as in older participants (Table 4). The increases in F1 + 2 (Figure 1C) and of TAT (Figure 1D) were similar in both groups of participants.

### 3.7. Effects of Orthostatic Challenge on Markers of Blood Vessel Damage in Young vs. Older Participants

Orthostatic challenge did not affect plasma levels of both tPA and TF in the younger participants (Table 3) as well as in the older participants (Table 4).

### 3.8. Effects of Orthostatic Challenge on Haematocrit in Young vs. Older Participants

Orthostatic challenge was associated with haemoconcentration in both young and older participants. Haematocrit was significantly higher in the post-standing samples of young (Table 3) and older participants (Table 4) as compared with the respective baseline levels. The increases in haematocrit were approximately the same in both groups of participants. The percent changes in plasma volume (calculated from haematocrit, according to van Beaumont [25]) was approximately 4.3% in the younger and 4.8% in older participants; no differences between the groups were, however, seen. Platelet counts were, commensurable with the haemoconcentration, increased by orthostatic challenge in both young (Table 3) and older participants (Table 4) to approximately the same extent.

## 4. Discussion

Our study demonstrates that older participants at rest are not hypercoagulable as compared with younger healthy participants. TEM-derived coagulation times as well as prothrombin times were prolonged, TF levels lower and t-PA plasma levels higher in the older vs. the young. Moreover, the procoagulant challenge of a simple sit-to-stand test shifted the haemostatic system of the younger, but not that of the older participants, towards a hypercoagulable state. 

Arterial or venous thrombosis primarily afflicts older individuals [26,27]. A hypercoagulable state in older persons has also been generally presumed [4]. We investigated whether the haemostatic system of older participants is, as widely reported, in a prethrombotic state. We, therefore, compared various coagulation parameters in the plasma and whole blood in older participants with those of younger healthy participants. Moreover, we investigated whether the haemostatic system of older participants can be more easily shifted towards a hypercoagulable state than that of the younger participants during a simple sit-to-stand test (orthostatic challenge). This test has previously been shown to stimulate the coagulation system and endothelium (although only mildly) and can be used to identify persons with an elevated risk of thrombosis [15].

Three of the CAT-derived resting parameters showed an elevated capability to generate thrombin in older participants as compared with the younger ones. This is in good agreement with several other studies showing significantly increased thrombin generation with aging [28,29,30,31]. We speculate that the elevated capacity of older participants in our study to generate thrombin could be partially attributable to the accompanying low plasma levels of PS present. We observed that baseline plasma levels of PS in older participants were approximately 20% lower than in the younger participants. It has previously been reported that dose-dependent increases in thrombin generation occur in the presence of lower PS plasma concentrations [32,33]. Indeed, in agreement with our findings, a slight reduction in PS levels with increasing age has also been shown by Dykes et al. [34]. On the other hand, some studies have reported similar plasma levels of PS in persons of different ages [35]. Moreover, mechanisms other than low levels of PS might be responsible for the efficient thrombin generation in older participants measured via CAT, since this technique is of limited sensitivity towards the PC/PS system [36].

In contrast to the differences seen across the ages in the resting thrombin generation parameters, most of the measured coagulation parameters did not support the assumption that (healthy) older participants are in a hypercoagulable state. In fact, even a hypocoagulable state was seen in the older participants. For example, TEM-derived coagulation times (CT) as well as prothrombin times (PT) were significantly prolonged in older as compared to younger participants, suggesting even a decelerated clot formation in older participants. Furthermore, our data show that baseline levels of tPA were markedly higher in the older participants, indicating enhanced fibrinolytic activity in plasma from older participants [37] and, moreover, TF levels, the physiological trigger of the coagulation cascade, were significantly lower in older participants. Overall, our data also do not support the widely held assumption that older persons are in a prethrombotic state.

In addition to the partially surprising results obtained in the resting plasma samples, the sit-to-stand test did not support the hypothesis that older healthy participants exhibit a hypercoagulable state during standing up. In contrast, orthostatic challenge in the younger participants was associated with a significantly higher capability to generate thrombin post-standing as compared to resting/baseline plasma samples (reflected in significantly increased endogenous thrombin potentials (ETPs) post-standing). This effect was not observed in the older participants. Whereas only ETP was altered by orthostatic challenge in the young participants, all other thrombin generation parameters (LT, Peak, ttPeak, VELINDEX, StartTail) remained unaffected. FVIII plasma levels were increased by orthostatic challenge in the young participants but not in the older participants. Elevated FVIII levels have been reported to support thrombin formation [38].

In addition, the effects of orthostatic challenge on thrombelastometry parameters and standard coagulation times PT and APTT do not suggest hypercoagulability in older participants. Thrombelastometry values were unchanged in young and in older participants, APTTs were shortened to approximately the same extent in both age groups and PTs were shortened to a greater extent in younger participants as compared to in older participants. In addition, standing-induced increases in F1 + 2 and TAT (post-standing vs. baseline) were similar in both age groups. These findings are not suggestive of hypercoagulability during standing up in older healthy participants. Notably, orthostatic challenge was associated with a slight but significant haemoconcentration. The haematocrit changes were approximately the same in younger and older participants (~2%), indicating comparable microvascular permeability in both age groups.

Generally speaking, our data suggest that older persons are not in a hypercoagulable state neither before nor after the application of a sit-to-stand test (believed to be potentially procoagulant in nature) as compared to younger persons. Our findings are even more surprising because our older study participants had higher BMIs and a higher proportion of them had arterial hypertension and/or dyslipidaemia.

Pathological conditions, such as malignancy, major surgery or autoimmune disease favour an activation process that is detected in blood by the presence of certain activation markers; these markers include products which are generated during the activation of coagulation factors or release products of activated thrombocytes and endothelial cells, respectively. Increased expression of tissue factor (TF) in monocytes is found in major surgery [39]. Moreover, elevated levels of TF and TF-positive extracellular vesicles have been found in cancer patients [40].

Numerous studies have reported that several factors involved in coagulation tip towards hypercoagulability with increasing age. For example, increased platelet activity [7,41], increased von Willebrand factor levels [42,43] and increased levels of fibrinogen, FV, FVIII and FIX [44,45] have been reported. However, the results from our study suggest that all these age-related changes are not potent enough to shift the coagulation system of older healthy persons towards hypercoagulability. Based on our results, we suggest that the increased thrombosis propensity clinically observed in older persons is not primarily attributable to a hyperactive coagulation cascade but rather to numerous other pathological conditions related to aging. For instance, stiffening and dilation of arteries due to an increase in collagen content [46], changes in the vascular content of heparin sulphate which regulates the anticoagulant efficacy of antithrombin [47] and/or increasing levels of homocysteine which induce oxidative stress in vascular endothelial cells [48].

Usually, hospitalised and immobilised patients, particularly at higher age, are routinely treated with anticoagulants. The results from our study suggest that only older patients with an existing higher risk for thrombosis (for example, due to major surgery or malignant disease) should undergo anticoagulant treatment; old age per se does not justify anticoagulant treatment. This is in agreement with the evidence-based clinical practice guidelines by the American College of Chest Physicians stating that “for acutely ill hospitalized medical patients at low risk for thrombosis, we recommend against the use of pharmacologic prophylaxis or mechanical prophylaxis” [49].

A limitation of the present study is the relatively low number of participants included. In order to generate clinically relevant data, a future study including a significantly higher number of participants has to be performed, comparing the haemostatic system across the ages. 

Therefore, a future study must be performed comparing the haemostatic system at baseline and after procoagulant challenge in a considerably higher number of young and older participants.

## 5. Conclusions

We did not find a hyperactive coagulation cascade in the older participants. The increased thrombosis propensity in older subjects might be attributable to other pathological conditions related to aging.

## Figures and Tables

**Figure 1 biomedicines-10-02769-f001:**
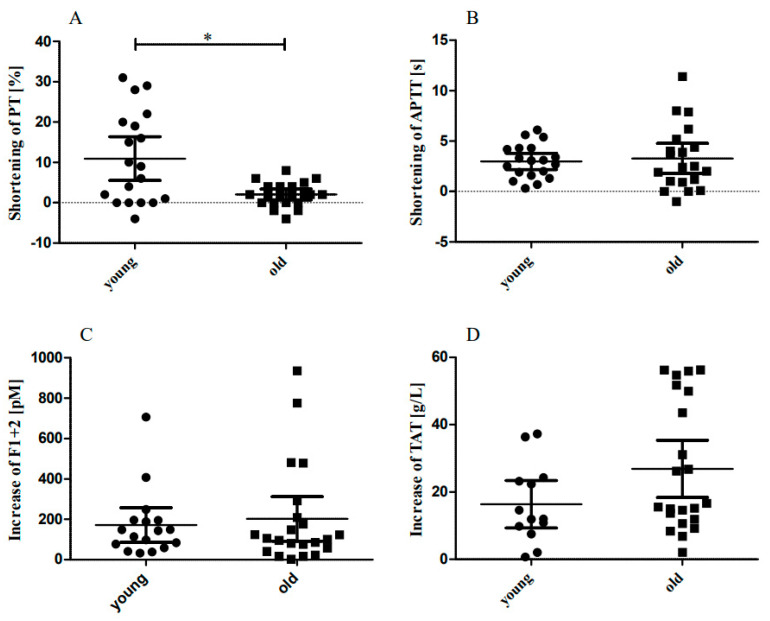
Coagulation activation due to orthostatic challenge. The sit-to-stand test caused coagulation activation in both young and older participants. Panel (**A**): the shortening of prothrombin times (PT) was significantly higher in young compared with older participants (*p* = 0.0242). Panel (**B**): the shortening of activated partial thromboplastin times (APTTs) was similar in young and older participants. Panel (**C**): the increases of Prothrombin fragment 1 + 2 (F1 + 2) were similar in young and older participants. Panel (**D**): the increases in Thrombin/antithrombin complexes (TAT) were similar in young and older participants. * *p* < 0.05.

**Table 1 biomedicines-10-02769-t001:** Participants’ characteristics. In total, 20 young and 22 older participants performed a sit-to-stand test. ASA, acetylsalicylic acid; NIHSS, National Institute of Health Stroke Scale; NOAC, non-vitamin K antagonist oral anticoagulant; OAC, oral anticoagulation. * *p*-values were calculated by means of the Mann–Whitney test.

	Young (n = 20)	Older (n = 22)
Gender, female/male	9/11	10/12
Age (years), median ± 95% CI * Body mass index, median ± 95% CI, kg/m^2^ *	22.0 (21.2–24.3) 22.5 (21.7–25.3)	61.0 (59.4–65.3) 27.6 (25.4–29.7)
Antiplatelet/anticoagulant therapy		
ASA (100 mg), n (%)	0 (0)	3 (14)
P2Y12-inhibitors, n (%)	0 (0)	0 (0)
NOAC/OAC, n (%)	0 (0)	0 (0)
Antidiabetic therapy, n (%)	0 (0)	1 (5)
Antihypertensive therapy, n (%)	0 (0)	7 (32)
Previous vascular events, n (%)	0 (0)	0 (0)
Vascular risk factors		
Arterial hypertension, n (%) Dyslipidemia, n (%) Diabetes mellitus, n (%) Nicotine abuse Obesity > grade 1, n (%) Atrial fibrillation, n (%)	0 (0) 0 (0) 0 (0) 1 (5) 0 (0) 0 (0)	6 (27) 5 (23) 1 (5) 0 (0) 1 (5) 0 (0)
NIHSS at baseline	0	0

**Table 2 biomedicines-10-02769-t002:** Coagulation values prior to orthostatic challenge. Baseline levels of coagulation values in young (n = 20) and older (n = 22) participants. Data are expressed as median ± 95% CI. *p*-values were calculated by means of the Mann–Whitney test.

	Young	Older	*p*-Value
Thrombelastometry (TEM)			
Coagulation time (CT, s) Clot formation time (CFT, s) Maximum clot firmness (MCF, mm) Alpha angle (°)	209 (196–235) 179 (147–184) 55.0 (53.1–57.3) 58.0 (55.7–62.2)	295 (248–330) 154 (134–213) 57.5 (53.8–60.4) 60.0 (54.8–64.4)	**0.0025** 0.6068 0.1926 0.5504
Calibrated automated thrombogram (CAT)			
Lag time (LT, min) Endogenous thrombin potential (ETP, nM·min) Peak (nM) Time to peak (ttPeak, min) VELINDEX (nM/min) StartTail (min)	2.7 (2.4–2.8) 1337 (1258–1619) 216 (193–275) 6.6 (5.9–7.1) 59.0 (47.1–87.6) 22.5 (21.2–23.6)	2.7 (2.6–3.2) 1554 (1398–1645) 281 (275–318) 5.7 (5.3–6.1) 106.8 (95.7–120.5) 21.2 (20.5–22.1)	0.3223 0.2100 **0.0004** **0.0333** **0.0004** 0.2311
Standard coagulation parameters			
Prothrombin time (PT, %) Activated partial thromboplastin time (APTT, s) Factor II (FII, %) Factor VII (FVII, %) Factor VIII (FVIII, %) Protein C (PC, %) Protein S, (PS, %)	114 (106–128) 34.4 (32.9–36.6) 106 (104–117) 100 (88–111) 111 (98–127) 105 (102–142) 112 (98–135)	103 (98–109) 37.2 (34.8–39.2) 119 (103–125) 113 (92–130) 100 (37–151) 109 (100–120) 92 (79–100)	**0.0087** 0.0868 0.4744 0.2393 0.2476 0.4783 **0.0176**
Thrombin generation			
Prothrombin fragment 1 + 2 (F1 + 2, pM) Thrombin/antithrombin complexes (TAT, g/L)	162 (153–288) 4.5 (3.7–5.0)	266 (246–325) 4.0 (2.6–6.8)	**0.0044** 0.1244
Blood vessel damage			
Tissue plasminogen activator (tPA, ng/mL) Tissue factor (TF, pg/mL)	3.8 (3.1–5.2) 672 (596–779)	7.3 (6.6–10.4) 440 (413–666)	**0.0002** **0.0245**
Haematocrit (Hct, %) Platelet count (10^3^/mL)	39.5 (37.9–41.0) 244 (218–254)	38.0 (36.5–39.5) 212 (189–224)	0.1744 0.1083

**Table 3 biomedicines-10-02769-t003:** Effect of orthostatic challenge on coagulation values in young participants (n = 20). Data are expressed as median ± 95% CI. *p*-values were calculated by means of the Wilcoxon matched pairs signed-rank test.

	Baseline	Post-Standing	*p*-Value
Thrombelastometry (TEM)			
Coagulation time (CT, s) Clot formation time (CFT, s) Maximum clot firmness (MCF, mm) Alpha angle (°)	209 (196–235) 179 (147–184) 55.0 (53.1–57.3) 58.0 (55.7–62.2)	201 (183–224) 156 (141–178) 54.0 (52.9–56.8) 61.0 (57.7–63.2)	0.1633 0.3165 0.9300 0.2756
Calibrated automated thrombogram (CAT)			
Lag time (LT, min) Endogenous thrombin potential (ETP, nM·min) Peak (nM) Time to peak (ttPeak, min) VELINDEX (nM/min) StartTail (min)	2.7 (2.4–2.8) 1337 (1258–1619) 216 (193–275) 6.6 (5.9–7.1) 59.0 (47.1–87.6) 22.5 (21.2–23.6)	2.7 (2.5–2.9) 1350 (1304–1645) 227 (199–276) 6.6 (6.0–7.1) 54.2 (49.2–89.6) 22.7 (21.5–24.0)	0.2693 **0.0264** 0.6474 0.4764 0.4859 0.1400
Standard coagulation parameters			
Prothrombin time (PT, %) Activated partial thromboplastin time (APTT, s) Factor II (FII, %) Factor VII (FVII, %) Factor VIII (FVIII, %) Protein C (PC, %) Protein S, (PS, %)	114 (106–128) 34.4 (32.9–36.6) 106 (104–117) 100 (88–111) 111 (98–127) 105 (102–142) 112 (98–135)	120 (115–140) 31.1 (30.1–33.4) 110 (107–121) 100 (91–113) 123 (120–150) 118 (108–139) 106 (102–135)	**0.0015** **0.0001** 0.0517 0.0971 **0.0003** 0.8752 0.4844
Thrombin generation			
Prothrombin fragment 1 + 2 (F1 + 2, pM) Thrombin/antithrombin complexes (TAT, g/L)	162 (153–288) 4.5 (3.7–5.0)	297 (249–530) 17.8 (13.5–30.0)	**0.0003** **0.0002**
Blood vessel damage			
Tissue plasminogen activator (tPA, ng/mL) Tissue factor (TF, pg/mL)	3.8 (3.1–5.2) 672 (596–779)	3.9 (3.1–5.4) 668 (603–785)	0.1701 0.9039
Haematocrit (Hct, %) Platelet count (10^3^/mL)	39.5 (37.9–41.0) 244 (218–254)	40.5 (39.1–42.0) 246 (223–262)	**0.0015** **0.0215**

**Table 4 biomedicines-10-02769-t004:** Effect of orthostatic challenge on coagulation values in older participants (n = 22). Data are expressed as median ± 95% CI. *p*-values were calculated by means of the Wilcoxon matched pairs signed-rank test.

	Baseline	Post-Standing	*p*-Value
Thrombelastometry (TEM)			
Coagulation time (CT, s) Clot formation time (CFT, s) Maximum clot firmness (MCF, mm) Alpha angle (°)	295 (248–330) 154 (134–213) 57.5 (53.8–60,4) 60.0 (54.8–64.4)	280 (242–336) 153 (140–245) 56.0 (53.2–59.2) 61.5 (54.7–64.3)	0.7938 0.8666 0.3940 0.8403
Calibrated automated thrombogram (CAT)			
Lag time (min) Endogenous thrombin potential (ETP, nM·min) Peak (nM) Time to peak (ttPeak, min) VELINDEX (nM/min) StartTail (min)	2.7 (2.6–3.2) 1554 (1398–1645) 281 (275–318) 5.7 (5.3–6.1) 106.8 (95.7–120.5) 21.2 (20.5–22.1)	2.9 (2.6–3.2) 1448 (1411–1668) 293 (269–324) 5.8 (5.4–6.2) 105.8 (91.4–123.0) 21.5 (20.7–22.8)	0.6879 0.3135 0.2813 0.6134 0.7151 0.8496
Standard coagulation parameters			
Prothrombin time (PT, %) Activated partial thromboplastin time (APTT, s) Factor II (FII, %) Factor VII (FVII, %) Factor VIII (FVIII, %) Protein C (PC, %) Protein S, (PS, %)	103 (98–109) 37.2 (34.8–39.2) 119 (103–125) 113 (92–130) 100 (37–151) 109 (100–120) 92 (79–100)	105 (100–111) 32.9 (31.8–35.7) 124 (109–131) 114 (95–136) 120 (92–138) 107 (101–118) 87 (80–104)	**0.0086** **0.0004** 0.0630 0.1183 0.2500 0.4201 0.4138
Thrombin generation			
Prothrombin fragment 1 + 2 (F1 + 2, pM) Thrombin/antithrombin complexes (TAT, g/L)	266 (246–325) 4.0 (2.6–6.8)	413 (363–610) 24.4 (23.2–40.0)	**0.0001** **0.0001**
Blood vessel damage			
Tissue plasminogen activator (tPA, ng/mL) Tissue factor (TF, pg/mL)	7.3 (6.6–10.4) 440 (413–666)	9.7 (8.1–12.3) 485 (432–750)	0.0514 0.2586
Haematocrit (Hct, %) Platelet count (10^3^/mL)	38.0 (36.5–39.5) 212 (189–224)	39.8 (37.8–40.8) 215 (194–235)	**0.0025** **0.0381**

## Data Availability

The authors hereby declare that the data presented in this study will be presented upon request by the corresponding author.

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
