# Peer review of "Orthostatic Challenge-Induced Coagulation Activation in Young and Older Persons"

_biomedicines, 2022, doi:10.3390/biomedicines10112769_

Round 1
Reviewer 1 Report
In this study, the authors aimed to investigate whether orthostatic challenge induces changes in coagulation in older patients. Unfortunately, I found the findings of the study confusing, and thus the manuscript not suitable for publication. Another major issue, that the authors are aware of, is of course the small number of patients included.
According to the results of this study, orthostatic challenge has no effect on thromboelastometry values in younger or older patients. However, in an older study of the same group (reference number 15 of the manuscript), orthostatic challenge was associated with activation of the coagulation system and shifted the hemostatic system towards hypercoagulation in healthy older persons. Remarkably, the median/mean age of the ‘older’ persons was similar, 61 versus 65 years old.
Author Response
Author’s reply to Reviewer 1:
We completely agree with the reviewer that the number of patients is relatively small.
However, we do not think that the findings of the present study are contradictory to our previous study (Cvirn, G.; Kneihsl, M.; Rossmann, C.; Paar, M.; Gattringer, T.; Schlagenhauf, A.; Leschnik, B.; Koestenberger, M.; Tafeit, E.; Reibnegger, G.; et al. Orthostatic Challenge Shifts the Hemostatic System of Patients Recovered from Stroke toward Hypercoagulability. Front Physiol 2017, 8, 12, doi:10.3389/fphys.2017.00012.). In both studies we found activation of the coagulation system in response to orthostatic stress. Both thrombin/antithrombin complexes and prothrombin fragment 1+2 levels were markedly increased after orthostatic stress (in both studies). In our previous study thrombelastometry values were also not elevated after orthostatic stress in older patients. . Significantly shortened coagulation times in the previous study were found in patients recovered from stroke. However, this patient group was not investigated in the present study.
Moreover, the mean age of our older patients in the present study is not exactly the same as in our previous study. However, not exactly the same 22 older patients were included in the present study.
Reviewer 2 Report
The authors address the topic of Orthostatic chellenge-induced coagualation in younger and older people and attempt to implicate the outcomes of hypercoagulability testing in the rationale for thrombosis in the elderly.
They begin by noting that, in addition to the plasma coagulation system, platelets, the vascular wall and disturbances in fibrinolytic activity may also influence this condition. Other acquired factors are marginally discussed.
Addendum 1 Can the authors add the pathophysiology of thrombotic conditions in individual acquired conditions such as malignancy, major surgery, autoimmune disease, and also add the contribution of these conditions to thrombosis.
A very small sample size was selected for the patient population, which may be misleading, and the methodologies are described in a cursory manner, which may have implications for data interpretation.
Addition 2 Describe in detail all methodologies system method, exact reagents (e.g. for aPTT it is not clear whether it is sensitive to lupus anticoagulans as a thrombotic marker).
Addendum 3 For TGA, please describe in detail the sample preparation especially the double centrifugation with debris removal please document with validation of the procedure as ultrafiltration is used as standard.
Addendum 4 Please provide the age and validation of the KX-21 instrument.
The chapter on results is clearly structured
Question on Table 1 - Why are 3 propositions in the elderly group treated with ASA? Wouldn't it be appropriate to replace them with other proposites and thus stagnate nicotine abus in 1 proposite in the younger group.
Question for Table 2
Please provide statistics on when
CAT 216 vs. 281 p value is 0.0004 and t-PA 3.8 vs. 7.3 is 0.0002 these data are questionable
Question on Table 3
CAT 1337 vs.1350 p value is 0.0264 really significant ??? this data raises doubts and please document the pairwise plot.
Prothrombin time is normally issued in primary measured seconds and calibration of this test above 100% is not possible. If you have it please document the standard with a level of 120% . However, I believe the data is extrapolated. Please explain.
The results chapter from 3.3 to 3.8 contains a mixture of speculation and interpretation . Firstly it should be in the discussion chapter and secondly supported by the literature. Please add.
The discussion chapter is unbalanced and should include a comparison of the results against the literature. It needs to be substantially revised, keeping in mind the methodology of the thesis. Aim of the thesis, the set and its limitations , results and on the basis of these define the outcomes and discuss these.
Author Response
Author’s reply to Reviewer 2:
Ad 1) Pathological conditions, such as malignancy, major surgery, or autoimmue disease favour an activation process that is detected in blood by the presence of certain activation markers, these marker include products which are generated during the activation of coagulation factors or release products of activated thrombocytes and endothelial cells, respectively. Increased expression of tissue factor in monocytes is found in major surgery (Dahl OE. The role of the pulmonary circulation in the regulation of coagulation and fibrinolysis in relation to major surgery. J Cardiothorac Vasc Anesth. 1997 May;11(3):322-8. doi: 10.1016/s1053-0770(97)90102-6. PMID: 9161901 Review.). Moreover, elevated levels of tissue factor and tissue factor-positive extracellular vesicles have been found in cancer patients (Kim et al., Mechanisms and biomarkers of cancer-associated thrombosis. Transl Res. 2020 Nov;225:33-53. doi: 10.1016/j.trsl.2020.06.012). This is stated now on page 2, lines 58-64.
We completely agree with the reviewer that the number of participants is relatively small.
Ad 2) We added the exact reagents used for routine coagulation testing. We used Synthasil as aPTT reagent which is sensitive to lupus anticoagulant and want to point out, that none of the participants exhibited prolonged aPTT and that changes after orthostatic challenge are not likely to occur due to presence/absence of lupus anticoagulant.
It is stated now in the manuscript (page 4, lines 153-156): Determinations of PTs (expressed as % of normal), APTTs as well as of plasma activities of FII, FVII, FVIII, PC-, and PS-activity were performed on a ACL Top 350 (Werfen, Germany) using the PT reagent ReadiPlasTin, the aPTT reagent SynthASil, and HemosIL reagents for all specialized clotting assays (all from Werfen, Germany).
Ad 3) The PPP was centrifuged at 2600 x g immediately before it was used for CAT measurements. Then, plasma was carefully aliquoted to the 96-well plate leaving 100 µl at the bottom of the vial. We developed and validated these procedures years ago when we were studying the impact of microparticles on thrombin generation (Schweintzger et al. Thromb Res. 2011, Deutschmann et al. JPGN 2013). With our centrifugation scheme larger cell debris is removed, but the smaller fraction of microparticles remain in the sample. However, the method remains unaffected by microparticles when an excess of 4 µM phospholipids and 5 pM TF are added exogenously to the sample. This double centrifugation procedure has also been employed by HC Hemker who developed the assay. We refrained from ultrafiltration, because we observed substantial contact activation that would bias the results. This is stated now on page 3, lines 132-142.
Ad 4) The KX-21 was purchased in 2014 and validated in 2021.
Question on Table 1: The reviewer is completely right. However, due to the relatively low number of participants we did not want to reduce the number of participants even further.
Question on Table 2: Due to the low dispersion of the data, both Peak and t-PA values were significantly altered due to orthostatic stress. We applied the Mann-Whitney-Test, which is, in our opinion, the appropriate statistical test for this setting.
Question on Table 3: The reviewer is completely right, the difference between the medians is relatively small. However, the dispersion of data is relatively small and the Wilcoxon matched-pairs signed rank test is, in our opinion, appropriate for this setting.
Prothrombin times:
We have to apologize, the PT-values in [s] are not available. Following PTs [INR] were available:
|
Older, prior |
Older, after |
|
Young, prior |
Young, after |
|
0,92 |
0,92 |
|
1 |
0,9 |
|
0,83 |
0,83 |
|
0,9 |
0,8 |
|
0,98 |
0,97 |
|
0,9 |
0,88 |
|
0,97 |
0,95 |
|
0,8 |
0,7 |
|
0,96 |
0,97 |
|
0,9 |
0,9 |
|
0,94 |
0,94 |
|
1,1 |
1 |
|
0,96 |
0,92 |
|
0,9 |
0,9 |
|
1,05 |
1,04 |
|
0,88 |
0,87 |
|
0,98 |
0,95 |
|
1,1 |
1,05 |
|
0,91 |
0,89 |
|
0,79 |
0,8 |
|
0,87 |
0,86 |
|
0,8 |
0,75 |
|
1,04 |
1,06 |
|
0,95 |
0,88 |
|
0,99 |
0,98 |
|
0,8 |
0,71 |
|
0,98 |
0,99 |
|
0,9 |
0,82 |
|
1,08 |
1,07 |
|
0,9 |
0,9 |
|
0,97 |
0,94 |
|
0,9 |
0,89 |
|
0,92 |
0,91 |
|
0,87 |
0,87 |
|
0,91 |
0,91 |
|
0,75 |
0,72 |
|
0,92 |
0,91 |
|
0,99 |
0,98 |
|
0,98 |
0,95 |
|
|
|
|
0,99 |
0,97 |
|
|
|
In the discussion section we state that older healthy participants are not in a hypercoagulable state compared with young healthy participants. To the contrary, orthostatic challenge (a simple sit-to-stand test) caused a higher shifts towards hypercoagulability in young compared with older participants. We conclude that the increased thrombosis propensity in older persons is not attributable to a hyperactive coagulation cascade but to other pathologies associated with aging.
Round 2
Reviewer 1 Report
Lines 60-62 of the introduction, describing others' findings, would be better placed at discussion section, where further analysis could follow.
Please also rephrase line 350 of the conclusions ('..in the older ').
Author Response
Author’s reply to Reviewer 1:
As suggested by the reviewer, lines 60-62 are transferred from the Introduction section to the Discussion section (lines 330-336) now.
Line 350: Lines 350-353 are corrected now.
Reviewer 2 Report
The authors answered most of the questions I asked, except that I drew attention to possible discrepancies in the statistical evaluation of dta, when minimal changes, e.g. in table 4 Prothrombin time 103 vs. 105 or in Table 3 ETP 1337 vs. 1350 turned out to be statistically significant, which in my experience is not possible when the measurement uncertainty is taken into account. Please fill in the measurement uncertainties and recalculate the statistics.
Author Response
Author’s reply to Reviewer 2:
We completely agree with the reviewer that although the changes in the medians of ETP and PT were relatively small, statistical significance seems surprising. However, median values do not provide information on the individual trends of values compared to baselines, which was evaluated with paired-sample tests. Also, we minimized assay uncertainty by measuring paired samples within one run, thus, eliminating effects of inter-assay variability. In the tables below we provide information about the relative changes in ETP and PT, showing the change compared to baseline to clarify this issue.
|
Table 3 ETP (young) |
Baseline |
Post standing |
relative increase [%] |
|
|
1132,94 |
1218,59 |
7 |
|
|
1093,68 |
1138,09 |
4 |
|
|
1622,39 |
1691,24 |
4 |
|
|
1161,65 |
1314,05 |
13 |
|
|
1402,21 |
1386,61 |
-1 |
|
|
1556,02 |
1519,31 |
-2 |
|
|
1308,55 |
1279,95 |
-2 |
|
|
1205,37 |
1257,57 |
4 |
|
|
2381,19 |
2293,21 |
-3 |
|
|
1116,15 |
1171,31 |
5 |
|
|
1102,32 |
1091,95 |
0 |
|
|
1719,4 |
1748,96 |
2 |
|
|
1181 |
1210 |
2 |
|
|
1223 |
1237 |
1 |
|
|
1366 |
1450 |
6 |
|
|
1678 |
1701 |
1 |
|
|
1531 |
1680 |
10 |
|
|
2119 |
2152 |
2 |
|
Table 4, PT (older) |
Baseline |
Post standing |
Relative increase [%] |
|
|
111 |
111 |
0 |
|
|
134 |
134 |
0 |
|
|
99 |
101 |
2 |
|
|
101 |
105 |
4 |
|
|
103 |
101 |
-2 |
|
|
107 |
107 |
0 |
|
|
103 |
111 |
8 |
|
|
85 |
87 |
2 |
|
|
103 |
105 |
2 |
|
|
113 |
118 |
6 |
|
|
87 |
83 |
-3 |
|
|
97 |
99 |
2 |
|
|
99 |
97 |
-2 |
|
|
79 |
81 |
2 |
|
|
101 |
107 |
6 |
|
|
111 |
113 |
2 |
|
|
113 |
113 |
0 |
|
|
111 |
113 |
2 |
|
|
99 |
105 |
6 |
|
|
97 |
101 |
4 |
Round 3
Reviewer 2 Report
I thank the authors for providing the primary data that fully explain the measured results. At the moment, I have no doubts about the quality of the outputs, I would just point out to both the editors and the authors if it is not better to demonstrate this experiment on the difference in paired tests, where it would be completely obvious and unquestionably explained to the reader.